# Recombinant Actifensin and Defensin-d2 Induce Critical Changes in the Proteomes of Multidrug-Resistant *Pseudomonas aeruginosa* and *Candida albicans*

Ifeoluwa D. Gbala,[a] Rosaline W. Macharia,[b] Joel L. Bargul,[c,d] Gabriel Magoma[a,c]

aMolecular Biology and Biotechnology, Pan African University, Institute for Basic Sciences, Technology, and Innovation, Juja, Kenya
bDepartment of Biochemistry, University of Nairobi, Nairobi, Kenya
cJomo Kenyatta University of Agriculture and Technology, Juja, Kenya
dInternational Centre of Insect Physiology and Ecology, Nairobi, Kenya

**ABSTRACT** Drug-resistant strains of *Pseudomonas aeruginosa* and *Candida albicans* pose serious threats to human health because of their propensity to cause fatal infections. Defensin and defensin-like antimicrobial peptides (AMPs) are being explored as new lines of antimicrobials, due to their broad range of activity, low toxicity, and low pathogen resistance. Defensin-d2 and actifensin are AMPs from spinach and *Actinomyces ruminicola*, respectively, whose mechanisms of action are yet to be clearly elucidated. This study investigated the mechanisms of action of the recombinant AMPs through label-free quantitative proteomics. The data are available at PRIDE with accession number PXD034169. A total of 28 and 9 differentially expressed proteins (DEPs) were identified in the treated *P. aeruginosa* and *C. albicans*, respectively, with a 2-fold change threshold and *P* values of <0.05. Functional analysis revealed that the DEPs were involved in DNA replication and repair, translation, and membrane transport in *P. aeruginosa*, while they were related mainly to oxidative phosphorylation, RNA degradation, and energy metabolism in *C. albicans*. Protein-protein interactions showed that the DEPs formed linear or interdependent complexes with one another, indicative of functional interaction. Subcellular localization indicated that the majority of DEPs were cytoplasmic proteins in *P. aeruginosa*, while they were of nuclear or mitochondrial origin in *C. albicans*. These results show that recombinant defensin-d2 and actifensin can elicit complex multiple organism responses that cause cell death in *P. aeruginosa* and *C. albicans*.

**IMPORTANCE** AMPs are considered essential alternatives to conventional antimicrobials because of their broad-spectrum efficacy and low potential for resistance by target cells. In this study, we established that the recombinant AMPs defensin-d2 and actifensin exert proteomic changes in *P. aeruginosa* and *C. albicans* within 1 h after treatment. We also found that the DEPs in peptide-treated *P. aeruginosa* are related to ion transport and homeostasis, molecular functions including nucleic and amino acid metabolism, and structural biogenesis and activity, while the DEPs in treated *C. albicans* are mainly involved in membrane synthesis and mitochondrial metabolism. Our results also highlight ATP synthase as a potential drug target for multidrug-resistant *P. aeruginosa* and *C. albicans*.

**KEYWORDS** *Pseudomonas aeruginosa*, *Candida albicans*, antimicrobial peptides, spinach defensin, actifensin, proteomics, antimicrobial mechanism

*P*seudomonas aeruginosa is a Gram-negative, nonfermenting, and metabolically versatile pathogen that is capable of adaptation and survival in various niches (1). Classes of antibiotics such as β-lactams, polymyxins, cephalosporins, and aminoglycosides are used to treat *P. aeruginosa* infections (2); however, drug-resistant strains of

Address correspondence to Ifeoluwa D. Gbala, ifeoluwa.gbala@students.jkuat.ac.ke.
The authors declare no conflict of interest.

*P. aeruginosa* exhibit extensive drug resistance to these classes of antibiotics (3), thus making it imperative to develop effective therapies against drug-resistant *P. aeruginosa*. Similarly, *Candida albicans* can cause serious infections in immunocompromised patients and is recognized as a major agent of nosocomial infections (4). In recent years, there has been a marked increase in the incidence of treatment failures in candidiasis cases due to drug resistance (5). Population-based studies have estimated that candidemia affects more than 250,000 persons worldwide every year, leading to more than 50,000 deaths (6).

Nonconventional antimicrobials such as phages and antimicrobial peptides (AMPs) are currently being explored as pharmacologically important alternatives to combat the menace of antimicrobial resistance (7). AMPs represent a part of the innate immune system in almost all classes of life (8) and have been reported to exhibit multiple drug targets simultaneously, making them less prone to resistance (9). Defensin-d2 is a cationic cysteine-rich plant defensin that was isolated from the leaves of *Spinacia oleracea* (spinach), which is a widely distributed leafy vegetable renowned for its nutritional benefits and antimicrobial activity (10–12). Defensin-d2 has been reported to be active against phytopathogens, including *Pseudomonas syringae*, *Clavibacter michiganensis*, *Ralstonia solanacearum*, and *Fusarium culmorum* (10–12), but little is known regarding its activity against human pathogens. Further, actifensin is a novel bacteriocin produced by *Actinomyces ruminicola*, which, like plant defensins, is cysteine-rich and contains disulfide bonds (13). It is reported to show remarkable antibacterial activity against Gram-positive bacteria, including methicillin-resistant *Staphylococcus aureus* (13), but its anticandidal activity has not been established.

Investigating the protein profile changes of pathogens in response to antimicrobial treatment is crucial in obtaining a global overview of the potential mechanisms of action of antimicrobial candidates (14). Studies have shown that a key antibacterial or antifungal mechanism of action of AMPs is through membrane permeability or disruption of cell wall synthesis, which leads to cell damage (15, 16). It has been suggested, however, that AMPs can also affect a range of molecular targets necessary for cell growth and viability, such as nucleic acids, enzymes, and other essential proteins (17–19). Previous studies compared the differential expression levels of proteins of *P. aeruginosa* and *C. albicans* in response to antimicrobial exposure using mass spectrometry as a reliable technique, because of its good repeatability, accurate quantitation, and identification of a wide range of proteins (18, 19).

In an earlier study (20), we expressed defensin-d2 and actifensin as recombinant peptides and determined their MICs against *P. aeruginosa* and *C. albicans*. The study further established that both peptides exhibited membrane permeabilization activities and induced reactive oxygen species (ROS) production in *P. aeruginosa* and *C. albicans*. To further gain comprehensive insight into the possible mechanisms of actions of these two peptides, this study used liquid chromatography-electrospray ionization-tandem mass spectrometry (LC-ESI-MS/MS) to determine the changes in protein profiles of multidrug-resistant *P. aeruginosa* and *C. albicans* strains following exposure to recombinant defensin-d2 and actifensin. Further, we report on the protein-protein interactions and functional annotations of the proteins.

## RESULTS AND DISCUSSION

**Treated pathogens expressed distinct protein profiles.** With a screening criterion of >2-fold change in abundance, a total of 28 and 9 proteins were found to be differentially expressed in treated *P. aeruginosa* and *C. albicans* samples, respectively. Among these differentially expressed proteins (DEPs), 10 (71.4%) proteins were upregulated in actifensin-treated *P. aeruginosa* (APA) and 4 (28.6 %) were downregulated (Table 1). Conversely, 10 (55.6%) proteins were downregulated and 8 (44.4%) were upregulated in defensin-treated *P. aeruginosa* (DPA). Five (83.3%) of the DEPs were downregulated in defensin-treated *C. albicans* (DCA), while 4 (80%) were downregulated in actifensin-treated *C. albicans* (ACA) (Table 2).

**TABLE 1** Overview of the DEPs in actifensin- and defensin-treated *P. aeruginosa*

| Protein no. | Protein identification (UniProtKB database identifier/ GenBank accession number/locus name) | Description | Change for[a]: | |
|---|---|---|---|---|
| | | | DPA | APA |
| 1 | sp/O52759/RS6_PSEAE | 30S ribosomal protein S6 | Up | Non |
| 2 | sp/P08280/RECA_PSEAE | Protein RecA | Down | Non |
| 3 | sp/P09591/EFTU_PSEAE | Elongation factor Tu | Up | Non |
| 4 | sp/P38100/CARB_PSEAE | Carbamoyl-phosphate synthase large chain | Down | Down |
| 5 | sp/P48247/GSA_PSEAE | Glutamate-1-semialdehyde 2,1-aminomutase | Down | Non |
| 6 | sp/Q51390/GLPK2_PSEAE | Glycerol kinase 2 | Down | Non |
| 7 | sp/Q51561/RPOB_PSEAE | DNA-directed RNA polymerase $\beta$ subunit | Up | Non |
| 8 | sp/Q9HT18/ATPA_PSEAE | ATP synthase $\alpha$ subunit | Down | Non |
| 9 | sp/Q9HUW9/Y4841_PSEAE | Uncharacterized Nudix hydrolase PA4841 | Up | Up |
| 10 | sp/Q9HV43/DNAK_PSEAE | Chaperone protein DnaK | Down | Non |
| 11 | sp/Q9HWC6/RL1_PSEAE | 50S ribosomal protein L1 | Up | Up |
| 12 | sp/Q9HWE1/RS5_PSEAE | 30S ribosomal protein S3 | Up | Non |
| 13 | sp/Q9I244/EFG2_PSEAE | Elongation factor G2 | Down | Non |
| 14 | sp/Q9I3F5/ACNA_PSEAE | Aconitate hydratase A | Down | Non |
| 15 | sp/Q9I467/COBQ_PSEAE | Cobyric acid synthase | Down | Non |
| 16 | tr/AAG07261/G3XD87_PSEAE | Respiratory nitrate reductase $\beta$ chain | Down | Non |
| 17 | tr/Q9HUY5/Q9HUY5_PSEAE | Magnesium-transporting ATPase, P-type 1 | Up | Up |
| 18 | tr/Q9HY79/Q9HY79_PSEAE | Bacterioferritin | Up | Up |
| 19 | tr/Q9I157/Q9I157_PSEAE | PvdL | Up | Non |
| 20 | sp/P53593/SUCC_PSEAE | Succinate-CoA ligase (ADP-forming) $\beta$ subunit | Non | Down |
| 21 | sp/Q9HWG0/UVRA_PSEAE | UvrABC system protein A | Non | Up |
| 22 | sp/Q9I5Y4/PGK_PSEAE | Phosphoglycerate kinase | Non | Up |
| 23 | sp/Q9I788/EXOT_PSEAE | Exoenzyme T | Non | Up |
| 24 | tr/Q9HY55/Q9HY55_PSEAE | Phosphoenolpyruvate-protein phosphotransferase | Non | Up |
| 25 | tr/Q9HZR3/Q9HZR3_PSEAE | CFTR inhibitory factor, Cif | Non | Down |
| 26 | tr/Q9I2T7/Q9I2T7_PSEAE | Probable ATP-binding component of ABC transporter | Non | Down |
| 27 | tr/Q9I2W9/Q9I2W9_PSEAE | Phosphoenolpyruvate synthase | Non | Up |
| 28 | tr/Q9I6K7/Q9I6K7_PSEAE | Sulfate-binding protein | Non | Up |

[a]Non, protein without significant up- or downregulation.

Notably, the peptides resulted in distinct expression profiles for proteins in both *P. aeruginosa* and *C. albicans*. Uniquely, all of the treated samples, except APA, exhibited downregulation of ATP synthase F1 $\alpha$ subunit. ATP synthases are membrane-bound enzyme complexes or ion transporters that utilize ATP hydrolysis for the transport of protons across a membrane (21). Thus, a functional ATP synthase that has been validated as a drug target (22) is essential for maintaining viability and metabolic propensity in bacterial and fungal pathogens (23). In *P. aeruginosa*, multiple studies have elucidated ATP synthase as essential for growth and pathogenicity under different growth conditions (24, 25). Similarly, ATP synthase has been reported to be crucial for maintaining *C. albicans* pathogenicity by assisting carbon flexibility (26). Specifically, carbamoyl-phosphate synthase was significantly downregulated in both APA and DPA, while four proteins (Nudix hydrolase, 50S ribosomal protein L1, magnesium-transporting P-type 1 ATPase, and bacterioferritin) were upregulated with both treatments. In all living organisms,

**TABLE 2** Overview of the DEPs in actifensin- and defensin-treated *C. albicans*

| Protein no. | Protein identification (UniProtKB database identifier/ GenBank accession number/locus name) | Description | Change for[a]: | |
|---|---|---|---|---|
| | | | DCA | ACA |
| 1 | tr/KAF6070929/A0A8H6C3V0_CANAX | 3-Demethylubiquinone-9-3-*O*-methyltransferase | Down | Non |
| 2 | tr/KAF6063297/A0A8H6BW35_CANAX | ATP synthase F1 $\alpha$ subunit | Down | Down |
| 3 | tr/Q9P841/Q9P841_CANAX | Galactose/glucose transporter | Down | Non |
| 4 | tr/Q0ZIF4/Q0ZIF4_CANAX | ATP synthase $\beta$ subunit | Down | Down |
| 5 | tr/EEQ41957/C4YG33_CANAW | MMS5_N domain-containing protein | Down | Non |
| 6 | tr/EEQ44166/C4YPM7_CANAW | Uncharacterized protein | Up | Non |
| 7 | tr/KAF6072181/A0A8H6C4W7_CANAX | CEK family protein | Non | Up |
| 8 | tr/KAF6070020/A0A8H6F3I7_CANAX | Poly(A) polymerase head domain family protein | Non | Down |
| 9 | tr/EEQ43918/C4YMT7_CANAW | Protein kinase domain-containing protein | Non | Down |

[a]Non, protein without significant up- or downregulation.

carbamoyl-phosphate, produced by carbamoyl-phosphate synthase, is a precursor for the synthesis of arginine and pyrimidines, which are essential in amino acid and nucleic acid synthesis, and thus affects physiological and biochemical functions (27). Studies demonstrated that disruption of carbamoyl-phosphate synthase in *Xanthomonas citri* and *Pseudomonas syringae* resulted in loss of pathogenicity, reduced motility, and attenuated biofilm formation (28, 29). The upregulated proteins mentioned above have overlapping functions; the magnesium-transporting P-type ATPase is important for $Mg^{2+}$ import into the cytoplasm in order to maintain the homeostasis needed for ribosome stability, prevent nitro-oxidative stress, and function as a cofactor for enzymatic reactions, such as hydrolysis of pyrophosphates carried out by Nudix hydrolase and iron metabolism by bacterioferritin (30–32). Therefore, the upregulation of these proteins can be attributed to internal regulation of the treated *P. aeruginosa* to protect itself from the stress induced by exposure to recombinant actifensin and defensn-d2 (32, 33).

ATP synthase $\alpha$ and $\beta$ subunits were significantly downregulated in both ACA and DCA, while ACA showed upregulation of choline/ethanolamine kinases (CEKs). Particularly in *C. albicans*, ATP synthase is crucial to maintaining cell viability, carbon metabolism, and pathogenicity (26). A recent study reported that inhibition of the $F_1F_0$-ATP synthase $\beta$ subunit could be responsible for *C. albicans* infection failure by disrupting carbon flexibility, which supports the proliferation of *C. albicans* in lipid- and amino-acid-rich microenvironments (34). We suggest that the ability of both recombinant peptides to reduce cell viability of *C. albicans* drastically, as seen in this study, could be strongly attributed to the ATP synthase inhibitory potentials of the peptides. The upregulation of CEKs, which are responsible for synthesis of phosphatidylcholine during phospholipid metabolism, in ACA is indicative of the hyperactivation of a regulatory pathway in *C. albicans* to maintain structural integrity in the presence of the stressor. Phospholipids (present as phosphatidylcholine or phosphatidylethanolamine) are the major structural lipids that form cellular membranes in *C. albicans* (35); therefore, it is necessary for *C. albicans* to synthesize them as precursors to maintain viability and support growth. In addition to their structural role within the cell, phospholipids may function as regulatory components (36, 37).

**Functional profiling and enrichment analysis of DEPs.** The DEPs in the treated samples were further analyzed for Gene Ontology (GO) annotation, Clusters of Orthologous Genes (COG) classification, KEGG pathway, enrichment, protein-protein interactions, and subcellular localizations.

**(i) GO annotation of DEPs in *P. aeruginosa*.** The changes in the protein profile of *P. aeruginosa* after recombinant actifensin exposure were mostly observed in the proteins involved in molecular function and biological processes, specifically binding (50% upregulated and 21.4% downregulated), catalytic activity (42.9% upregulated and 35.7% downregulated), cellular processes (35.7% upregulated and 21.4% downregulated), and metabolic processes (28.6% upregulated and 21.4% downregulated) (Fig. 1a). Also affected are proteins that function in cell parts, cellular component organization or biogenesis, transporter activity, biological regulation, response to stimulus, and localization. Defensin-d2 treatment of *P. aeruginosa* (Fig. 1b) also induced changes in the expression of proteins involved in catalytic activity (26.3% upregulated and 47.4% downregulated), binding (42.1% upregulated and 42.1% downregulated), cellular processes (36.8% upregulated and 47.4% downregulated), metabolic processes (21.1% upregulated and 42.1% downregulated), cell parts (15.8% upregulated and 26.3% downregulated), and response to stimulus (21.1% downregulated). Other proteins affected are associated with biological regulation, macromolecular complexes, structural molecular activity, cellular component organization, or biogenesis and localization.

Whether simultaneously or sequentially, we established that both recombinant peptides were able to affect proteins involved in different gene processes, thus strongly suggesting multiple mechanisms of action of the recombinant peptides against *P. aeruginosa*. While we acknowledge the structural or membrane-targeted action of our recombinant AMPs against *P. aeruginosa* in our study, similar to other reports on cationic AMPs (38–41), we postulate that it is not the sole mechanism of action exhibited by the two AMPs. A more likely mechanism of action of the recombinant peptides against *P. aeruginosa* is inhibition of

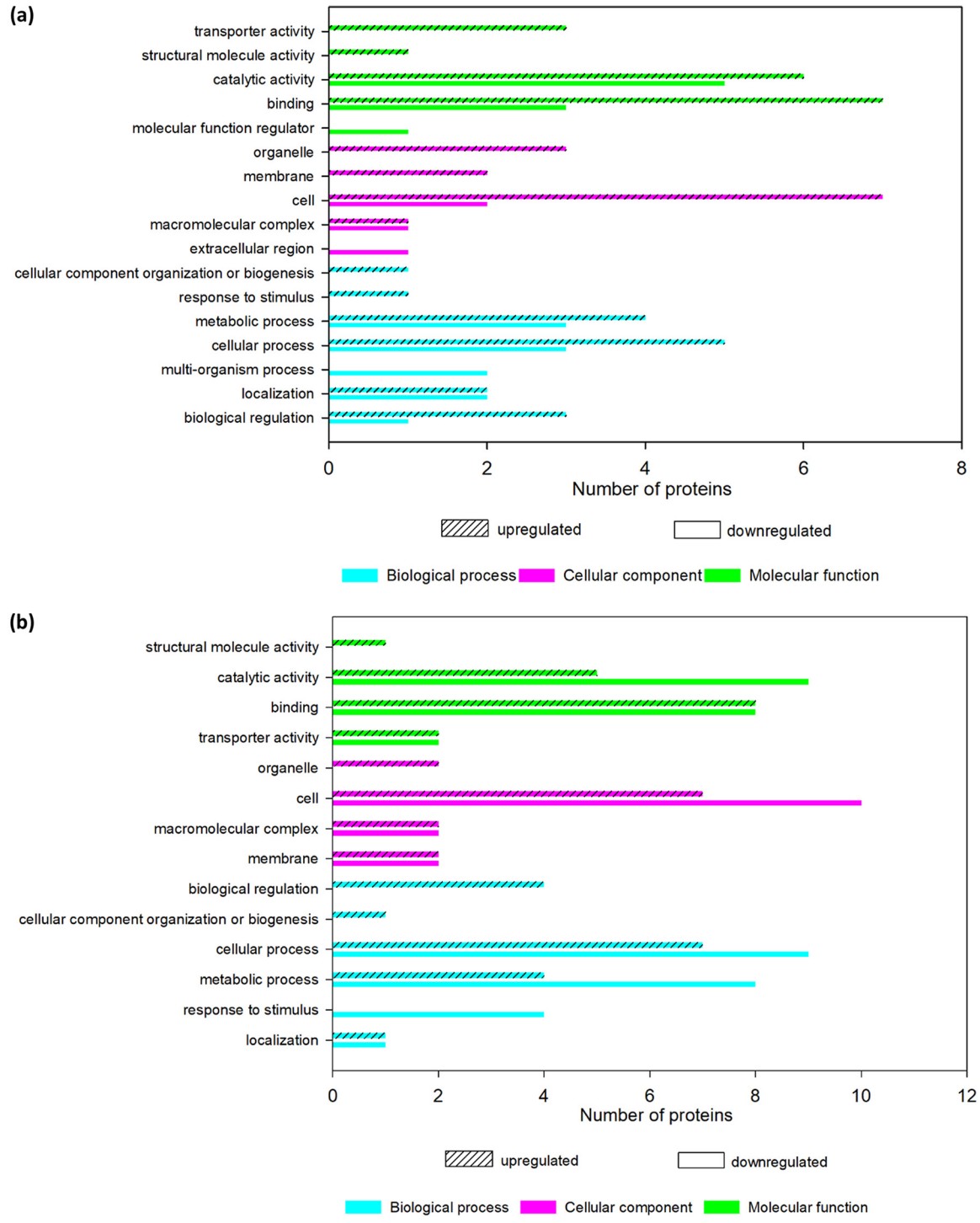

FIG 1 GO annotation of the DEPs in treated *P. aeruginosa*. (a) Annotation for APA. (b) Annotation for DPA.

molecular functions of the organism through interference with nucleic acid and/or protein synthesis after membrane disruption. Existing antibiotics classified as molecular function inhibitors target the DNA/RNA polymerase (42), ATP-dependent kinases (43), ATP synthase (23), or ribosomes (44, 45), similar to the observations of our study.

**(ii) GO annotation of DEPs in *C. albicans*.** The changes in the protein profile of *C. albicans* after recombinant actifensin exposure were mostly observed in the proteins

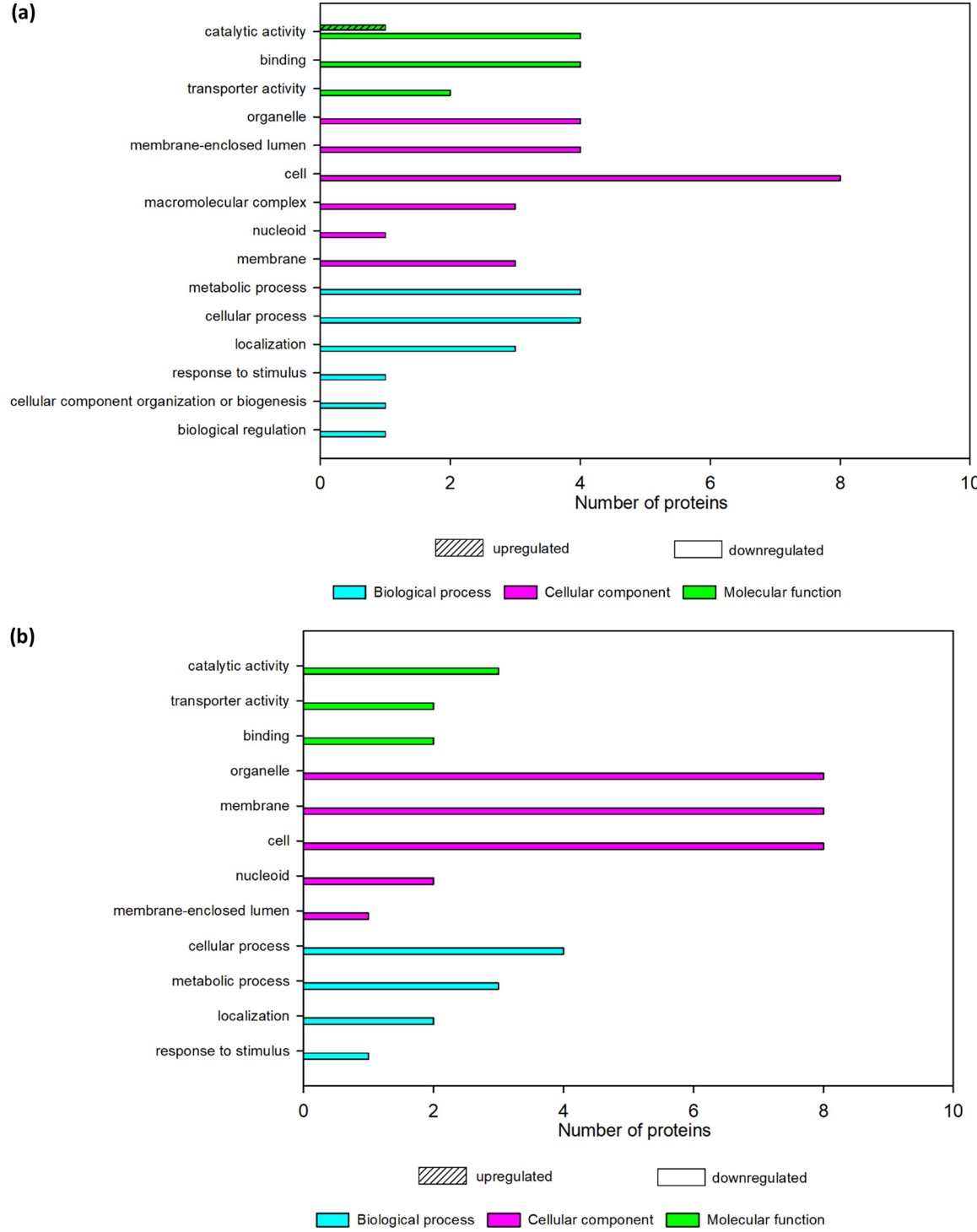

**FIG 2** GO annotation of the DEPs in treated *C. albicans*. (a) Annotation for ACA. (b) Annotation for DCA.

involved in cellular components, specifically in organelle parts (80% downregulated), membrane-enclosed lumen (80% downregulated), cell parts (80% downregulated), and macromolecular complexes (60% downregulated) (Fig. 2a). Also strongly affected were proteins that function in catalytic activity (20% upregulated and 80% downregulated), binding (80% downregulated), and metabolic and cellular processes (80% downregulated). Other proteins associated with the response to stimulus, transporter

activity, nucleoid, and localization, among others, were also downregulated. Similar to ACA, exposure of *C. albicans* to defensin-d2 treatment resulted in pronounced down-regulation of proteins associated with cellular components (organelle, membrane, cell, nucleoid, and membrane-enclosed lumen) (Fig. 2b).

Based on this result, we suggest the mechanism of action of the recombinant AMPs against *C. albicans* as mainly membrane targeted. We postulate that the disruption of membrane integrity or membrane permeability in the treated organism induced dys-functional transmembrane transport, resulting in the increased leakage of ATP and accumulation of oxidative stress, which in turn altered nucleotide synthesis, affected the mitochondria, or caused DNA/RNA degradation. This correlates with our previous study that showed that both peptides induced oxidative stress in *C. albicans* within 1 h after exposure (20). Other studies (46–48) also reported a membrane-targeted mecha-nism of action of AMPs against *C. albicans*. Yang et al. (19) also reported that DEPs in *C. albicans* treated with AMP-17 for 12 h were closely related to cell wall synthesis, RNA degradation, and oxidative stress. Cellular damage in *C. albicans* has been reported to cause excessive accumulation of ROS, which may lead to oxidative damage of nucleic acids, proteins, and lipids (46) and mitochondrial dysfunction characterized by loss of ATP (49).

**Pathway annotation for DEPs in *P. aeruginosa* shows high levels of enrichment of nucleotide excision and repair and ABC transporter pathways.** The most enriched pathways for DEPs in APA include ABC transporter, fructose and mannose metabolism, terpenoid backbone biosynthesis, nucleotide excision repair, and the phosphotransferase system (Fig. 3a). Upregulated proteins were mainly involved in energy metabolism, DNA replication and repair, and membrane transport, while the downregulated proteins were involved in pathogenesis, membrane transport, and nucleotide and amino acid metabo-lism. In DPA (Fig. 3b), proteins involved in pathways for porphyrin, pyrimidine, and nitro-gen metabolism were downregulated, while those involved in ribosome and terpenoid backbone synthesis pathways were upregulated. These pathway annotations of the DEPs strongly corroborate the action of the peptides on the membrane and molecular functions of *P. aeruginosa* largely by affecting the metabolic pathways, especially nucleic acid and amino acid metabolism. Yasir et al. (40) also reported the effect of cationic peptides on DNA after membrane disruption in a fluorescent-dye-based experiment. Therefore, the multiple pathways annotated for the DEPs further reiterate that the recombinant peptides can exert multiple mechanisms of action against *P. aeruginosa*.

**Pathway annotation for DEPs in *C. albicans* shows high levels of enrichment of oxidative phosphorylation and cell cycle pathways.** In both ACA and DCA, the oxi-dative phosphorylation and cell cycle (yeast meiosis) pathways were highly enriched. Both treatments also significantly downregulated RNA transport, starch and sucrose metabolism, and biosynthesis of secondary metabolites. Downregulated DEPs in both ACA and DCA were involved in cell growth, translation, and energy and carbohydrate metabolism, while lipid metabolism was upregulated (Fig. 4a and b). The upregulation of proteins involved in lipid metabolism can be attributed to internal regulation of *C. albicans* to repair and maintain membrane integrity. This is because lipids are crucial constituents of the membranes and they regulate cell proliferation, viability, and, in the case of pathogenic strains, virulence (50). Thus, the pathways affected by recombi-nant actifensin and defensin-d2 in *C. albicans* are important metabolic pathways for ATP production, cellular respiration and growth, carbon metabolism, and membrane synthesis. These findings suggest that both peptides exert a strong membrane-disrupt-ing antifungal action on *C. albicans*. Yang et al. (19) also reported significant enrich-ment of oxidative phosphorylation and RNA degradation in *C. albicans* treated with AMP-17.

**Protein-protein interactions show interdependence of membrane integrity and molecular functions.** Protein-protein interactions are highly specific contacts established between proteins because of biochemical events steered by their interactions (51). In APA, three of the nodes representing upregulated proteins clustered together, i.e., phosphoe-nolpyruvate phosphotransferase, phosphoglycerate kinase, and magnesium-transporting

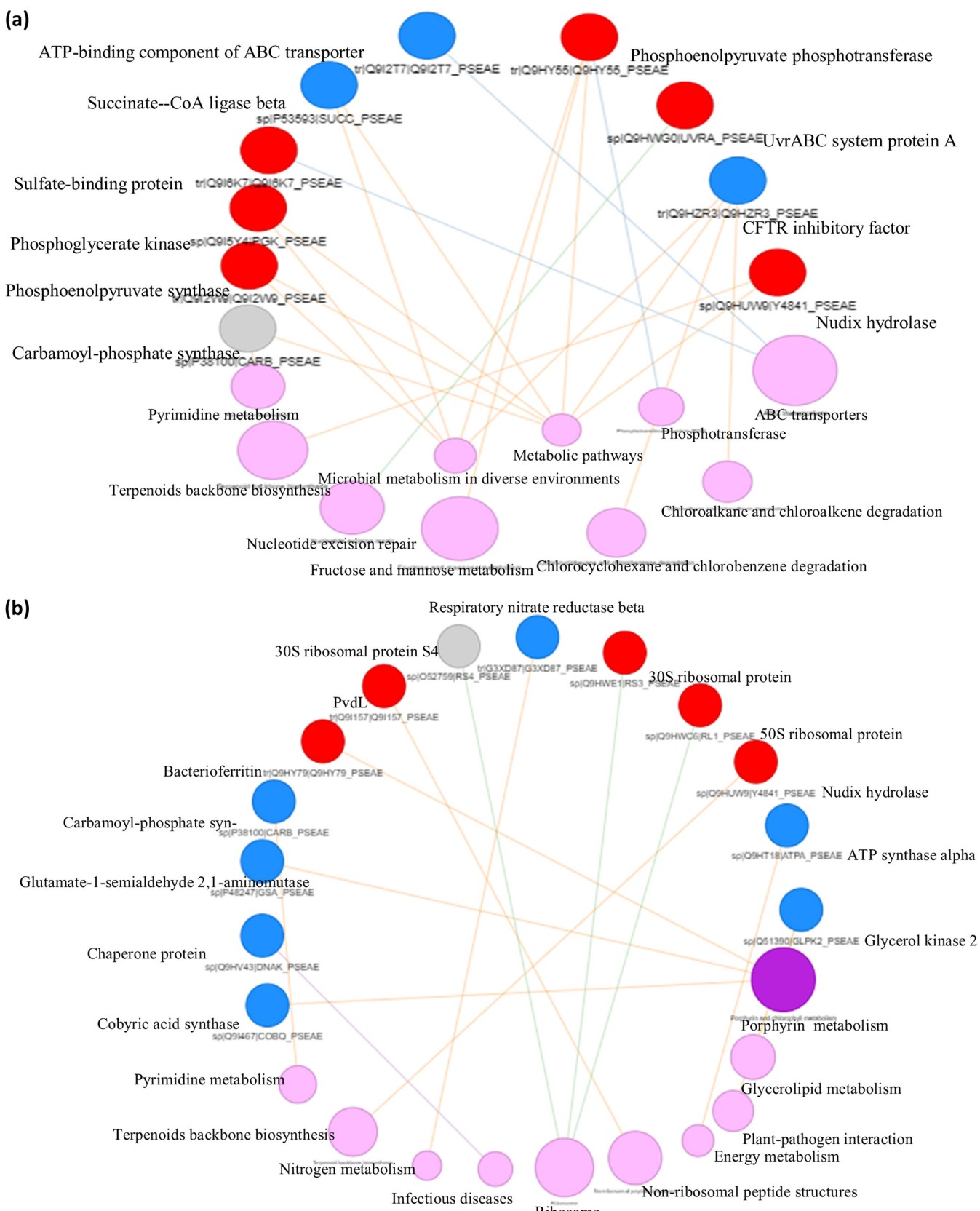

**FIG 3** Pathway annotation of the DEPs in treated *P. aeruginosa*. (a) APA. (b) DPA. Purple balls represent the enriched pathways; the size and color gradient indicate the level of enrichment. The blue and red balls represent downregulation and upregulation, respectively.

**(a)**

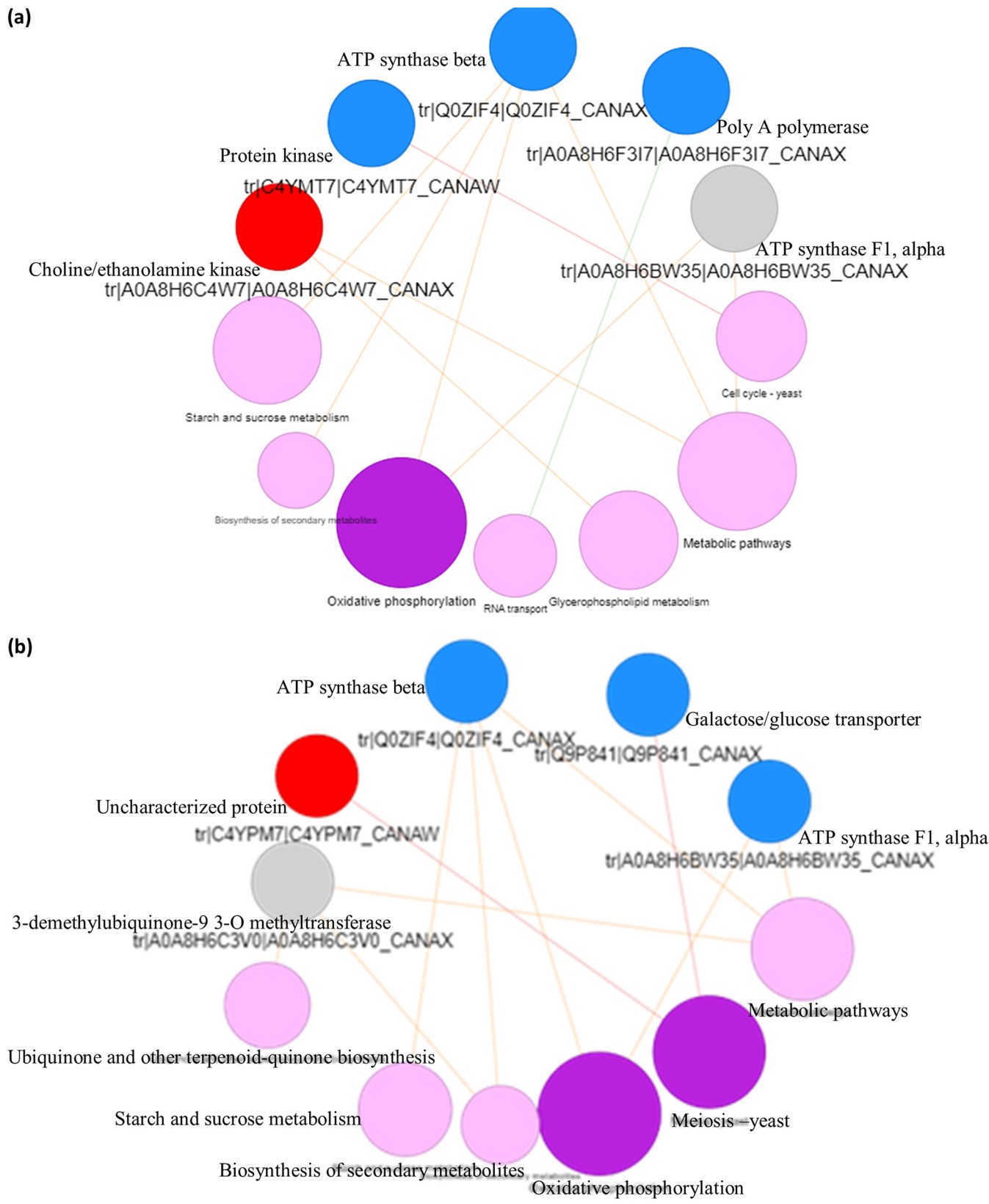

**(b)**

**FIG 4** Pathway annotation of the DEPs in treated *C. albicans*. (a) ACA. (b) DCA. Purple balls represent the enriched pathways; the size and color gradient indicate the level of enrichment. The blue and red balls represent downregulation and upregulation, respectively.

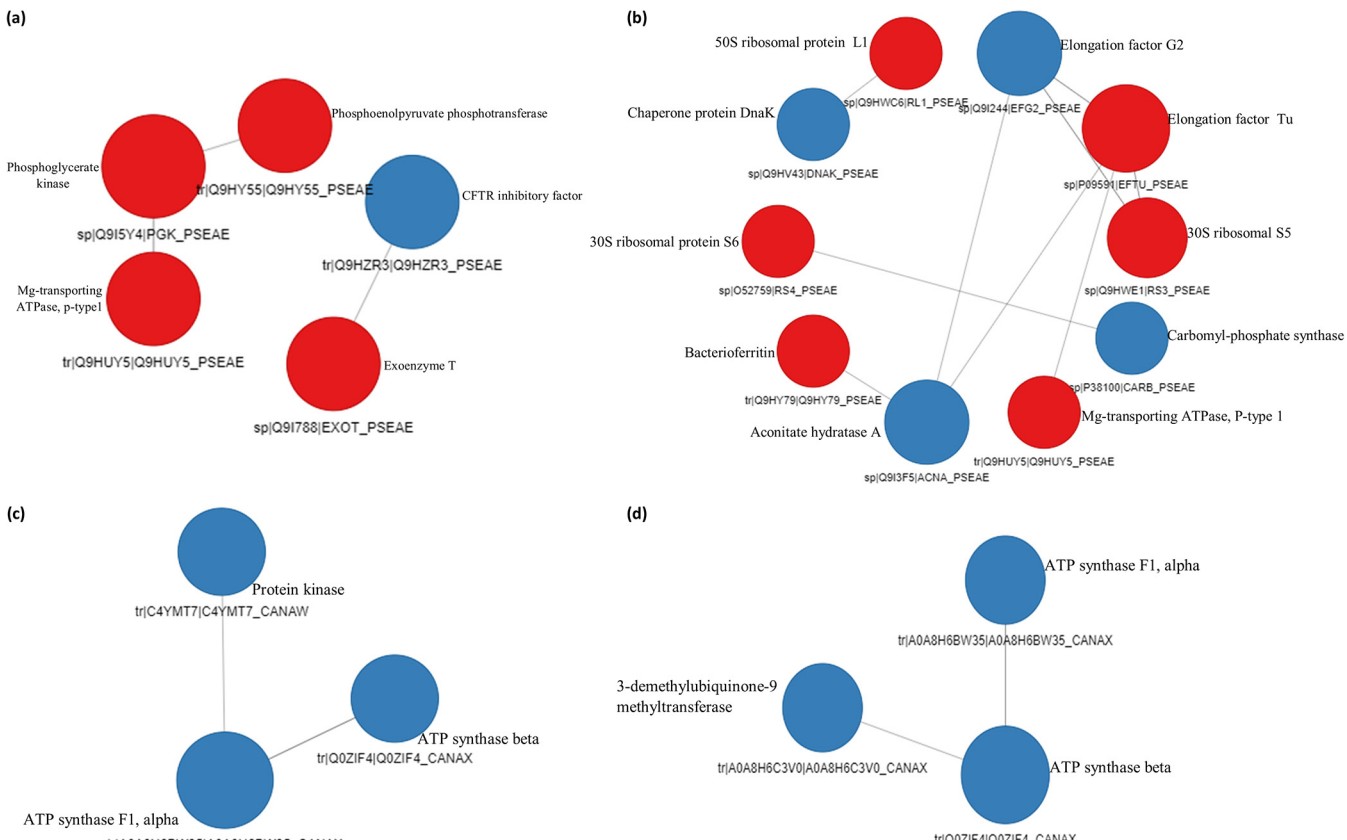

**FIG 5** Protein-protein interaction of DEPs in treated organisms. (a) APA. (b) DPA. (c) ACA. (d) DCA. The blue and red balls represent downregulation and upregulation, respectively.

P-type 1 ATPase. These proteins play major roles in sugar transport (52), gluconeogenesis (53), and cytoplasmic ion transport (54), respectively. These proteins are therefore jointly involved in a network of physiological and metabolic process regulation that is not limited to energy metabolism but also is involved in pathogenesis, interaction with nucleic acids, and cell viability (Fig. 5a). In DPA, a notable interdependent interaction was deduced for four nodes, representing elongation factor Tu (upregulated), 30S ribosomal protein S5 (up-regulated), elongation factor G2 (downregulated), and aconitate hydratase A (downregulated) (Fig. 5b). Downregulated chaperone protein DnaK, which plays a crucial role in DNA replication and repair and protein biogenesis (55), interacted linearly with 50S ribosomal protein L1 (upregulated), while the upregulated 30S ribosomal protein S6 interacted linearly with downregulated carbomyl-phosphate synthase large chain, an essential protein in amino acid metabolism. The up-down effects seen in the ribosomal and metabolic proteins present a scenario of overexpression of the ribosomal proteins (in order to synthesize essential proteins) in response to stress induced by the recombinant peptides.

In ACA, a linear path interaction was observed for three of the downregulated proteins, i.e., ATP synthase $\alpha$ subunit, ATP synthase $\beta$ subunit, and protein kinase domain-containing protein (Fig. 5c). In DCA, a linear interaction similar to that of ACA was obtained, with three nodes representing downregulated proteins ATP synthase $\alpha$ subunit, ATP synthase $\beta$ subunit, and 3-demethylubiquinone-9-3-O-methyltransferase (Fig. 5d). The interaction of ATP synthases, kinases, and transferase enzymes suggests that their metabolic actions are sequential to one another. The interaction is also in correlation with their functions in membrane ion transport and maintenance of cellular integrity (26, 34), further reiterating the membrane as a key target in the activity of the peptides against *C. albicans*.

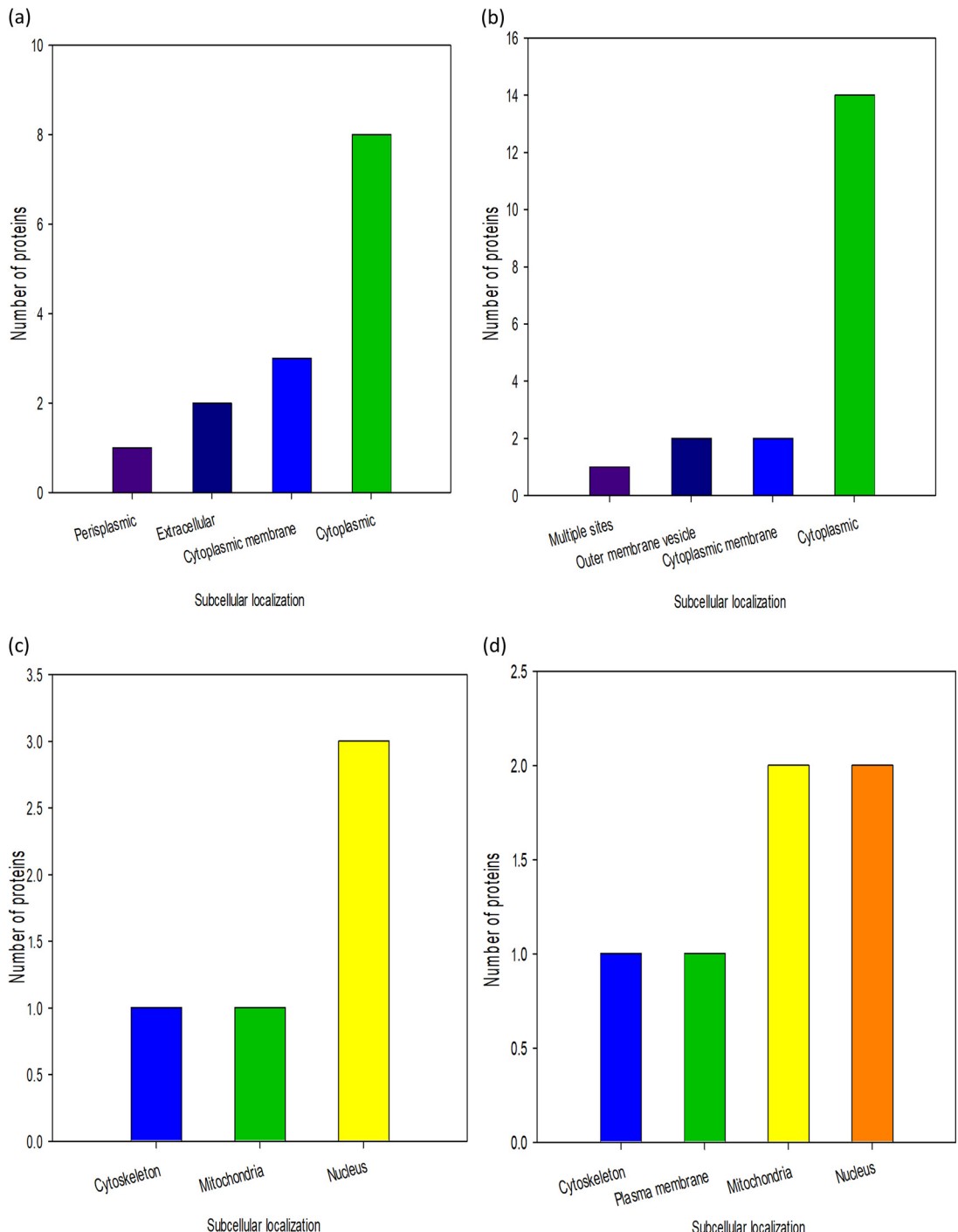

**FIG 6** Subcellular localization of DEPs in treated organisms. (a) APA. (b) DPA. (c) ACA. (d) DCA.

**Subcellular localization of DEPs.** The majority of the DEPs in APA and DPA were localized in the cytoplasm (57.1% and 73.7%, respectively). Other proteins were located in the cytoplasmic membrane, outer membrane vesicles, and extracellular and periplasmic membranes (Fig. 6a and b). This observation further affirms our postulation that membrane disruption is the initial step in the mechanism of action of both recombinant peptides against *P. aeruginosa*. In ACA and DCA, the DEPs were located in the nucleus (60% and 33.3%, respectively) and the mitochondria (20% and 33.3%, respectively). Other proteins were situated at the cytoskeleton and plasma membrane (Fig. 6c and d). These findings suggest that membrane permeability is a major mechanism of action of

the recombinant peptides against *C. albicans*, which resulted in effector actions seen in nuclear and mitochondrial damage.

We conclude that the recombinant peptides significantly affected the membrane proteins, ATP-dependent enzymes, and metabolic proteins involved in nucleic acid or amino acid synthesis in *P. aeruginosa*. A notable membrane-targeted action and excessive ROS production were also established as possible mechanisms of action of the recombinant peptides against *C. albicans*. We show that the recombinant peptides exerted multiple mechanisms of action against the test organisms, sequentially or simultaneously, based on the different pathways affected.

## MATERIALS AND METHODS

**Test organisms.** Cultures of *Pseudomonas aeruginosa* ATCC 27853 and *Candida albicans* ATCC 64124 were purchased from the American Type Culture Collection (ATCC). The propagation conditions for the isolates were as recommended by the ATCC, and isolates were preserved in Mueller-Hinton broth (MHB) at 4°C.

**Synthesis of recombinant peptides and determination of MICs.** The method of recombinant production of defensin-d2 and actifensin was described previously (20). Briefly, an overnight culture of transformed *Escherichia coli* Shuffle T7 (New England Biolabs, USA) containing recombinant plasmids (pTXB1-defensin-d2 and pTXB1-actifensin) was grown in modified Terrific broth (Sigma-Aldrich, Germany) containing 100 $\mu$g/mL ampicillin. Protein expression was induced with 0.4 mM isopropyl-$\beta$-D-thiogalactopyranoside (IPTG) at 30°C for 4 h. *E. coli* cells were lysed, and the clarified lysate was purified by chitin affinity chromatography. The MICs of recombinant actifensin against *C. albicans* and *P. aeruginosa* were 45 $\mu$g/mL and 1,448 $\mu$g/mL, respectively, while the MIC of recombinant defensin-d2 against *C. albicans* and *P. aeruginosa* was 7.5 $\mu$g/mL.

**Sample treatment.** Fresh colonies of *P. aeruginosa* and *C. albicans* grown on tryptic soy agar and Sabouraud dextrose agar, respectively, were subcultured in 5-mL of MHB. Inoculated MHB cultures were incubated at 37°C to an optical density at 600 nm ($OD_{600}$) of 0.1. Then, defensin-d2 and actifensin were added to the cultures to final concentrations equal to their respective MICs, for treatment. Untreated *P. aeruginosa* (UPA) and untreated *C. albicans* (UCA) were set up as controls. The tubes were gently mixed and further incubated at 37°C for 1 h.

**Protein extraction and digestion.** After the incubation period, *P. aeruginosa* and *C. albicans* cells were harvested using a centrifuge precooled to 4°C, at 5,000 rpm for 10 min. The cell pellets were washed using prechilled 1× phosphate-buffered saline (PBS) (pH 7.2) to remove all residual medium and resuspended in 600 $\mu$L of chilled TRIzol. The mixture was then incubated for 5 min at 24°C to lyse the cells and dissolve cell components. Subsequently, 600 $\mu$L of absolute ethanol was added to the suspension and thoroughly mixed to precipitate DNA. The mixture was loaded onto a Zymo-Spin II-CR column (Zymo Research, USA) and centrifuged at 11,000 × *g* for 30 s to remove RNA from the lysate. The flowthrough obtained was transferred on ice, and 1 mL of ice-cold acetone was added to 250 $\mu$L of the flowthrough and thoroughly mixed to precipitate the proteins in the mixture. The tubes were incubated on ice for 30 min and then centrifuged at 16,000 × *g* for 10 min. The supernatant was discarded, and the protein pellets obtained were washed with 400 $\mu$L of absolute ethanol by centrifugation at 16,000 × *g* for 1 min to remove residual solvent. The supernatant was discarded, and the pellets were air dried at room temperature. The air-dried pellets were resuspended in 8 M urea and stored at −80°C. The concentrations of the proteins were determined by the Bradford assay.

Then, 50 $\mu$g of each protein sample was diluted with 0.5 M triethylammonium bicarbonate to a final concentration of 1 M urea. Trypsin was then added to the protein samples in a 1:20 (wt/wt) ratio. The mixture was vortex-mixed briefly, centrifuged at 650 rpm for 1 min, and incubated at 37°C for 4 h. The digested peptides were freeze-dried using a lyophilizer.

**High-performance liquid chromatography.** The dried peptide samples were reconstituted with mobile phase A (2% acetonitrile, 0.1% formic acid) and centrifuged at 20,000 × *g* for 10 min, and the supernatant was taken for injection. Separation was performed with an UltiMate 3000 ultra-high-performance liquid chromatography (UHPLC) system (Thermo Fisher Scientific). The sample was first enriched in the trap column and desalted, and then it entered a self-packed $_{C18\ column}$ (internal diameter, 75 $\mu$m; column size, 3 $\mu$m; column length, 25 cm) and separated at a flow rate of 300 nL/min with the following effective gradient: 0 to 5 min, 5% mobile phase B (98% acetonitrile, 0.1% formic acid); 5 to 45 min, mobile phase B linearly increased from 5% to 25%; 45 to 50 min, mobile phase B increased from 25% to 35%; 50 to 52 min, mobile phase B increased from 35% to 80%; 52 to 54 min, 80% mobile phase B; 54 to 60 min, 5% mobile phase B.

**Detection of peptides by mass spectrometry.** The peptides separated by LC were ionized with a nano-ESI source and then passed to a tandem mass spectrometer (Q Exactive HF-X; Thermo Fisher Scientific, San Jose, CA) for data-dependent acquisition (DDA) mode detection. The main parameters were as follows: ion source voltage, 1.9 kV; MS5 scanning range, *m/z* 350 to 1,500 (resolution, 60,000); MS6 start, *m/z* 100 (resolution, 15,000). The ion-screening conditions for MS6 fragmentation were as follows: charge, +2 to +6; top 30 parent ions with peak intensity exceeding 10,000. The ion fragmentation mode was higher energy collisional dissociation (HCD), and the fragment ions were detected in the Orbitrap. The dynamic exclusion time was set to 30 s. The automatic gain control (AGC) was set as follows: MS5 3E6, MS6 1E5.

**Bioinformatics and statistical analysis of proteomic data.** The raw data were identified using the integrated Andromeda engine of MaxQuant v1.5.3.30. Further, MaxQuant was used to perform quantitative

analyses based on the peak intensity, peak area, and LC retention times of the peptides related to the first-order mass spectrometry. At the spectrum level, filtering was performed with a peptide spectrum match (PSM)-level false-discovery rate (FDR) of ≤1%; at the protein level, further filtering was performed with a protein-level FDR of ≤1% to obtain significant identification results. Parameters set in MaxQuant were as follows: fixed modifications, carbamidomethyl; variable modifications, oxidation (M), acetyl (protein N-term), deamidated (NQ), and Gln→pyro-Glu; peptide mass tolerance, 4.5 ppm; fragment mass tolerance, 20 ppm; minimal peptide length, 7 amino acids. The UniProt database (www.uniprot.org) (56) was used as the protein reference database for the analysis of both *P. aeruginosa* and *C. albicans* samples (untreated, defensin-treated, and actifensin-treated). The software was also used to extract peak areas and calculate protein quantitation values.

According to the set comparison groups (UPA versus APA, UPA versus DPA, UCA versus ACA, and UCA versus DCA), the multiples of differences in the proteins in each comparison group (untreated versus treated) were calculated using Welch's *t* test (57). Furthermore, screening to determine DEPs between the comparison groups was performed by setting fold change in the multiple of difference to >2. The resultant identified proteins in each sample were mapped to GO terms (http://www.geneontology.org) (58, 59). Functional category analysis was performed with Protein2GO and GO2Protein. KEGG (https://www.genome.jp/kegg/pathway.html) (60) was used for the pathway analysis of the identified proteins, to identify the important biochemical metabolic and signal transduction pathways of the proteins.

Pathway enrichment analyses were performed to compare the abundance of the specific terms or classification in the comparison groups with the natural abundance in the reference organism. Significant enrichment was set at *P* of <0.05. By comparison with the STRING protein interaction database (https://string-db.org) (61), the protein-protein interaction analysis was performed on the DEPs in the different comparison groups, and the interaction relationships were presented as a network map. Subcellular localization of the DEPs in the different comparison groups was also predicted using WoLF PSORT (https://wolfpsort.hgc.jp) (62) for *C. albicans* and PSORTb v3.0 (https://www.psort.org/psortb) (63) for *P. aeruginosa*.

**Data availability.** The mass spectrometry proteomic data have been deposited in the ProteomeXchange Consortium database via the PRIDE (https://www.ebi.ac.uk/pride/) (64) partner repository with the data set identifier PXD034169.

## ACKNOWLEDGMENT

This work was supported by the African Union Commission through the Pan African University scholarship.

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
