## [Reviewer comments · Microbiology Spectrum]

Microbiology Spectrum

Recombinant actifensin and defensin-d2 induce critical changes in the proteomes of multidrug-resistant *Pseudomonas aeruginosa* and *Candida albicans*

Ifeoluwa Gbala, ROSALINE MACHARIA, Joel Bargul, and Gabriel Magoma

Corresponding Author(s): Ifeoluwa Gbala, Pan African University, Institute for Basic Sciences, Technology and Innovation, Kenya

Review Timeline:

Submission Date:	June 3, 2022
Editorial Decision:	August 7, 2022
Revision Received:	August 12, 2022
Accepted:	September 6, 2022

Editor: Cezar Khursigara

Reviewer(s): Disclosure of reviewer identity is with reference to reviewer comments included in decision letter(s). The following individuals involved in review of your submission have agreed to reveal their identity: Xiangli Dang (Reviewer #1)

Transaction Report:

DOI: <https://doi.org/10.1128/spectrum.02062-22>

August 7, 2022

Mx. Ifeoluwa Deborah Gbala
Pan African University, Institute for Basic Sciences, Technology and Innovation, Kenya
Molecular Biology and Biotechnology
Jomo Kenyatta University of Agriculture and Technology, Nairobi, Kenya
Juja 62000
Kenya

Re: Spectrum02062-22 (Recombinant actifensin and defensin-d2 induce critical changes in the proteomes of multidrug-resistant *Pseudomonas aeruginosa* and *Candida albicans*)

Dear Mx. Ifeoluwa Deborah Gbala:

Two experts and I have reviewed your manuscript and agree that although the findings are interesting, modifications are required before it can be accepted for publication. Specifically, Review 2 has recommendations for validation that should be properly addressed. Please address all the reviewer's comments when submitting a revised version of your manuscript.

Link Not Available

Sincerely,

Cezar Khursigara

Journals Department
Reviewer comments:

Reviewer #1 (Comments for the Author):

The present study reported activity and mechanism of action of defensin-d2 and actifensin against *P. aeruginosa* and *C. albicans*. Major and minor comments to the authors are as follows:

1. Line 38 and 39, both peptides bactericidal and fungicidal effects within 1 hour of treatment. I did not find this assay.
2. Based on MIC of both peptides, defensin-d2 showed higher activity than actifensin. What's the difference of mode of action

between them?

3. *P. aeruginosa* and *C. albicans* contains more than 5000 and 6000 proteins, respectively. Why only 276 and 108 proteins was identified in this study, respectively?

4. In table 2, No 1 protein (30S ribosomal protein S6) also up regulated in DPA group, why not included in statistic analysis?

5. In Fig 1, 30S ribosomal protein S4 should be 30S ribosomal protein S6.

6. Line 178-181, based on Fig 1, seven of the downregulated proteins in both treatment showed close relatedness, but not five.

7. In Fig 1 and 2, why the annotation of protein fold change level is different?

8. Line 268, network actifensin-treated (b) should (d).

9. Please rebuild Fig 3 and 4 (c) and (d). It's difficult to understand the result.

10. In Fig 7, please indicate the red and blue meaning. Also, please indicate protein name, but not Protein ID.

11. Line 461, CFU should be given full name when first appears.

Reviewer #2 (Comments for the Author):

The manuscript describes the activity of two antimicrobial peptides against multi-drug resistant *Pseudomonas aeruginosa* and *Candida albicans*. Considering the great threat that antimicrobial resistance poses and that those two microorganisms are involved in a great number of infections caused by both of them, I think that the manuscript is interesting. However, I missed some extra experiments/data validating the findings from their proteomics analysis. The authors made an extensive analysis of the induced changes in the proteomic landscape in response to both peptides. That leads to several hypotheses regarding their possible mechanisms of action. However, the authors did not try to validate their findings using experimental approaches to investigate the effects of the peptides on both microbes. This would include well-established assays, to evaluate membrane or cell wall integrity, electron microscopy to evaluate morphological changes, or assays to evaluate the induction of oxidative stress after the treatments.

Omics assays are indeed very good to provide hypotheses, but I really think the authors should try to employ some other experimental approaches to validate their data.

In addition to that, the manuscript text is quite extensive in length, and I think it could be shortened to focus on the main changes that were observed.

They also need to increase the font size in the figures and I think that a better organization of the figures will improve the manuscript as well.

Staff Comments:

Preparing Revision Guidelines

Please return the manuscript within 60 days; if you cannot complete the modification within this time period, please contact me. If you do not wish to modify the manuscript and prefer to submit it to another journal, please notify me of your decision immediately so that the manuscript may be formally withdrawn from consideration by Microbiology Spectrum.

Response to Reviewer's Comments

Reviewer #1:

Comment 1: Line 38 and 39, both peptides bactericidal and fungicidal effects within 1 hour of treatment. I did not find this assay.

Response 1: these results and the methodology for the assay have been published in our earlier study. As such, to prevent dual publication, we have modified by citing the published work. Methodology and results are presented below for review purpose only:

Methodology

Broth microdilution method was employed to determine the minimum inhibitory concentrations (MIC) of the recombinant peptides against methicillin-resistant *S. aureus* (MRSA), *E. coli*, *K. pneumoniae*, *P. aeruginosa* and *C. albicans*. The cell density was adjusted to 0.5 McFarland standard (10^6 CFU/ mL) in normal saline, and *C. albicans* was further diluted to 10^3 CFU/ mL in Mueller Hinton broth (Oxoid, UK). Then, 10 μ L of the inoculum suspension was added into 100 μ L of concentrations of recombinant defensin (3.75 – 985 μ g/mL) or recombinant actifensin (11.5 – 2895 μ g/mL) diluted in Mueller Hinton broth. Ampicillin was used as positive control for *E. coli*, *K. pneumoniae*, *P. aeruginosa*; vancomycin for MRSA and nystatin for *C. albicans*. The plates were incubated at 37 °C for 24 h for bacteria or 48 h for *C. albicans*, then 30 μ L of resazurin (0.015%) was added to all wells and further incubated for 2 – 4 h to detect microbial activity by colour change from blue to pink. The experiments were performed in triplicates and the minimum inhibitory concentration was determined as the least concentration with no colour change for each organism. To determine the minimum bactericidal/fungicidal concentration (MBC/MFC), a loopful of inoculum from the wells without colour change were plated on Mueller Hinton agar. The plates were then incubated at 37 °C for 24 h. The lowest concentration that showed no colonies was taken as the MBC/MFC.

Minimum inhibitory/bactericidal/fungicidal concentrations of recombinant actifensin and defensin-d2

Test organism	MIC (μ g/mL)					MBC/MFC (μ g/mL)	
	Actifensin	Defensin-d2	Ampicillin	Nystatin	Vancomycin	Actifensin	Defensin-d2
MRSA	23	-	NA	NA	4	-	ND
E. coli	-	30	5000	NA	NA	ND	246
P. aeruginosa	1448	7.5	10000	NA	NA	1448	123
K. pneumoniae	-	30	5000	NA	NA	ND	-
C. albicans	23	7.5	NA	1290	NA	724	63

NA – antibiotics not applicable to organism; ND – Not determined; - indicates no MIC or MBC. Values presented are means of triplicate data and standard deviation is 0.00

Comment 2: Based on MIC of both peptides, defensin-d2 showed higher activity than actifensin. What's the difference of mode of action between them?

Response 2: from the earlier results of this study, we determined that defensin-d2 showed a broader antimicrobial spectrum compared to actifensin. We postulate that the differences in

the activity seen could be attributed to the origin of the antimicrobial peptides and the overall amino acid composition of the peptides. Plant defensins have been studied to exhibit broad-spectrum activity against pathogens while bacteriocins exhibit a narrow spectrum activity against bacteria especially Gram positive. Hence, in studies where bacteriocins showed a considerable broader activity, higher concentrations of the bacteriocins were used. While different concentrations are required to elicit cidal actions against *P. aeruginosa* and *C. albicans*, similar pattern of interactions are seen for both peptides.

Comment 3: *P. aeruginosa* and *C. albicans* contains more than 5000 and 6000 proteins, respectively. Why only 276 and 108 proteins was identified in this study, respectively?

Response 3: the number of total proteins identified in this study is attribute to the duration of growth of the test organisms as well as the data acquisition parameters used. Optimal growth is required for optimal protein expression, which in the case of both test organisms will require about 24 hours. But because our focus was in investigating the effect of the peptides at the onset of the log phase, obtaining the entire protein profile of the organisms was not a priority. We believe that the expressed proteins at this stage are very essential for the viability and essential metabolic activities of the organisms. Also, the top 30 parent ions with intensity over 10,000 were set as selection parameters for the dependent data acquisition in order to have representation of the most abundant proteins in the different samples.

Comment 4: In table 2, No 1 protein (30S ribosomal protein S6) also up regulated in DPA group, why not included in statistic analysis?

Response 4: This has been corrected.

Comment 5: In Fig 1, 30S ribosomal protein S4 should be 30S ribosomal protein S6

Response 5: this has been corrected.

Comment 6: Line 178-181, based on Fig 1, seven of the downregulated proteins in both treatment showed close relatedness, but not five.

Comment 7: In Fig 1 and 2, why the annotation of protein fold change level is different? The figures represent different microorganisms

Response to 6 & 7: the section on cluster analysis has been removed from the revise manuscript in order to reduce the length of the manuscript and focus on the highly significant findings. However, the annotations of the fold change were based on the highest values obtained for the different comparison groups.

Comment 8: Line 268, network actifensin-treated (b) should (d).

Response 8: this section has been modified.

Comment 9: Please rebuild Fig 3 and 4 (c) and (d). It's difficult to understand the result.

Response 9: the figures have been modified

Comment 10: In Fig 7, please indicate the red and blue meaning. Also, please indicate protein name, but not Protein ID.

Response 10: the figure has been modified.

Comment 11: Line 461, CFU should be given full name when first appears.

Response 11: this line has been excluded from the revised manuscript.

Reviewer #2:

The manuscript describes the activity of two antimicrobial peptides against multi-drug resistant *Pseudomonas aeruginosa* and *Candida albicans*. Considering the great threat that antimicrobial resistance poses and that those two microorganisms are involved in a great number of infections caused by both of them, I think that the manuscript is interesting. However, I missed some extra experiments/data validating the findings from their proteomics analysis. The authors made an extensive analysis of the induced changes in the proteomic landscape in response to both peptides. That leads to several hypotheses regarding their possible mechanisms of action. However, the authors did not try to validate their findings using experimental approaches to investigate the effects of the peptides on both microbes. This would include well-established assays, to evaluate membrane or cell wall integrity, electron microscopy to evaluate morphological changes, or assays to evaluate the induction of oxidative stress after the treatments. Omics assays are indeed very good to provide hypotheses, but I really think the authors should try to employ some other experimental approaches to validate their data. In addition to that, the manuscript text is quite extensive in length, and I think it could be shortened to focus on the main changes that were observed. They also need to increase the font size in the figures and I think that a better organization of the figures will improve the manuscript as well.

Response: In an earlier study published, we demonstrated membrane permeabilization of the peptides. We also determined the DNA-binding ability of the peptides by gel retardation assay and UV-Vis. Further we investigated the effects of the peptides on membrane-associated virulence factors. A few of the results are attached here for review purposes only.

‘Gbala ID, Macharia R, Bargul J, Magoma G. 2022. Membrane Permeabilization and Antimicrobial Activity of Recombinant Defensin-d2 and Actifensin against Multidrug-Resistant *Pseudomonas aeruginosa* and *Candida albicans*. *Molecules* 27(14): 4325.’

Figure. Relative fluorescence of PI uptake to assess plasma membrane permeability of recombinant defensin-d2 and actifensin against *P. aeruginosa* and *C. albicans*. Untreated samples were used for normalization. UPA – untreated *P. aeruginosa*; UCA – untreated *C. albicans*; def – defensin-d2; acti - actifensin

Figure. Relative fluorescence of 3-DiSc (5) uptake to assess inner membrane depolarization of recombinant defensin-d2 and actifensin against *P. aeruginosa* and *C. albicans*. 0.1% (v/v) Triton-X 100 was used for normalization of fluorescence. UPA – untreated *P. aeruginosa*; UCA – untreated *C. albicans*; def – defensin-d2; acti – actifensin.

Figure. Relative fluorescence of H₂DCFDA for assessing ROS production in *P. aeruginosa* and *C. albicans* treated with recombinant defensin-d2 and actifensin. 2mM H₂O₂ was for normalization of fluorescence. UPA – untreated *P. aeruginosa*; UCA – untreated *C. albicans*; def – defensin-d2; acti – actifensin.

Figure. Gel retardation analysis of *P. aeruginosa* and *C. albicans* genomic DNA treated with concentrations of defensin-d2.

Figure. Fold change in virulence gene expression of untreated and treated *P. aeruginosa* normalized to rpoS (a) and 16S rRNA (b).

Figure. Fold change in virulence gene expression of untreated and treated *C. albicans* normalized to actin (a) and 18S rRNA (b).

We are working on further validating the effects on genomic DNA as well as on transcription of virulence factors as well as genes involved in DNA replication and repair, and as such these data cannot be submitted as part of this manuscript at this time.

As the reviewer rightly mentioned, we agree that omics provides a global insight into possible mechanisms of action of antimicrobials. While our laboratory continues working to experimentally validate these postulates, we believe the content of this manuscript is highly important and sufficient in providing insights into the mechanisms of action of the peptides. We also believe this can generate research interest from other researchers.

Furthermore, the length of the manuscript as well as the figures have been modified to improve the manuscript.

September 6, 2022

Mx. Ifeoluwa Deborah Gbala
Pan African University, Institute for Basic Sciences, Technology and Innovation, Kenya
Molecular Biology and Biotechnology
Jomo Kenyatta University of Agriculture and Technology, Nairobi, Kenya
Juja 62000
Kenya

Re: Spectrum02062-22R1 (Recombinant actifensin and defensin-d2 induce critical changes in the proteomes of multidrug-resistant *Pseudomonas aeruginosa* and *Candida albicans*)

Dear Mx. Ifeoluwa Deborah Gbala:

Your manuscript has been accepted, and I am forwarding it to the ASM Journals Department for publication. You will be notified when your proofs are ready to be viewed.

Sincerely,

Cezar Khursigara
Editor, Microbiology Spectrum
